# Multi-Omics and Management of Follicular Carcinoma of the Thyroid

**DOI:** 10.3390/biomedicines11041217

**Published:** 2023-04-19

**Authors:** Thifhelimbilu Emmanuel Luvhengo, Ifongo Bombil, Arian Mokhtari, Maeyane Stephens Moeng, Demetra Demetriou, Claire Sanders, Zodwa Dlamini

**Affiliations:** 1Department of Surgery, Charlotte Maxeke Johannesburg Academic Hospital, University of the Witwatersrand, Parktown, Johannesburg 2193, South Africa; maeyane.moeng@wits.ac.za; 2Department of Surgery, Chris Hani Baragwanath Academic Hospital, University of the Witwatersrand, Johannesburg 1864, South Africa; ifongo.bombil@wits.ac.za; 3Department of Surgery, Dr. George Mukhari Academic Hospital, Sefako Makgatho Health Sciences University, Ga-Rankuwa 0208, South Africa; surgiderm@wol.co.za; 4SAMRC Precision Oncology Research Unit (PORU), DSI/NRF SARChI Chair in Precision Oncology and Cancer Prevention (POCP), Pan African Cancer Research Institute (PACRI), University of Pretoria, Hatfield 0028, South Africa; dd.demetriou@up.ac.za (D.D.); zodwa.dlamini@up.ac.za (Z.D.); 5Department of Surgery, Helen Joseph Hospital, University of the Witwatersrand, Auckland Park, Johannesburg 2006, South Africa; clairej.sanders@yahoo.com

**Keywords:** follicular carcinoma, genomics, multi-omics, pathomics, radiomics, transcriptomics, treatment

## Abstract

Follicular thyroid carcinoma (FTC) is the second most common cancer of the thyroid gland, accounting for up to 20% of all primary malignant tumors in iodine-replete areas. The diagnostic work-up, staging, risk stratification, management, and follow-up strategies in patients who have FTC are modeled after those of papillary thyroid carcinoma (PTC), even though FTC is more aggressive. FTC has a greater propensity for haematogenous metastasis than PTC. Furthermore, FTC is a phenotypically and genotypically heterogeneous disease. The diagnosis and identification of markers of an aggressive FTC depend on the expertise and thoroughness of pathologists during histopathological analysis. An untreated or metastatic FTC is likely to de-differentiate and become poorly differentiated or undifferentiated and resistant to standard treatment. While thyroid lobectomy is adequate for the treatment of selected patients who have low-risk FTC, it is not advisable for patients whose tumor is larger than 4 cm in diameter or has extensive extra-thyroidal extension. Lobectomy is also not adequate for tumors that have aggressive mutations. Although the prognosis for over 80% of PTC and FTC is good, nearly 20% of the tumors behave aggressively. The introduction of radiomics, pathomics, genomics, transcriptomics, metabolomics, and liquid biopsy have led to improvements in the understanding of tumorigenesis, progression, treatment response, and prognostication of thyroid cancer. The article reviews the challenges that are encountered during the diagnostic work-up, staging, risk stratification, management, and follow-up of patients who have FTC. How the application of multi-omics can strengthen decision-making during the management of follicular carcinoma is also discussed.

## 1. Introduction

Primary thyroid carcinoma (TC) can originate from the follicular cells, para-follicular cells, or lymphoid tissues. Thyroid cancers constitute 1–5% of malignancies in adults [1,2]. Cancers of the thyroid of follicular cell origin are divided into well-differentiated thyroid carcinoma (WDTC), poorly differentiated and undifferentiated/anaplastic carcinoma thyroid carcinoma (ATC) [1,2,3,4,5,6,7]. The WDTCs include papillary thyroid carcinoma (PTC), follicular thyroid carcinoma (FTC), and oncocytic cell carcinoma (OC) [8,9]. Papillary carcinoma and FTC consist of several subtypes. The diagnosis of PTC and FTC together with their subtypes is based on the presence of classical nuclear features and the architecture of the tumor [10,11,12].

Over 90% of WDTCs are sporadic. The major risk factors for WDTCs include previous exposure to ionizing radiation, a persistently abnormal level of iodine, and Hashimoto’s thyroiditis [13,14,15,16,17]. Around 85–90% of thyroid cancers in an iodine-replete environment are classical and other subtypes of PTC [18,19]. The prevalence of FTC is influenced by the iodine status of that region [20,21,22,23,24]. The experience of the histopathologists also influences the rate of diagnosis, as other benign and malignant lesions of the thyroid may be mistaken for FTC [25]. Although FTC can occur in children, it is predominantly a disease of females over the age of 40 [26,27].

The majority of patients who have TC present with euthyroid goiter, and a few present due to metastasis to cervical lymph nodes or distant sites from an occult primary tumor [28,29]. Sometimes a WDTC is detected incidentally during the histological analysis of a specimen following a thyroidectomy for supposedly benign goiter [30]. Sometimes, a patient who has a localized or metastatic FTC may present with hyperthyroidism [31,32]. The diagnostic work-up of a patient who is suspected to have TC includes thyroid function testing (TFT), ultrasound, and fine needle aspiration cytology (FNAC) [33,34]. The diagnosis of PTC following FNAC is based on the existence of typical nuclear features and/or architectural changes [35]. Supplementary imaging investigations, immunohistochemistry, mutational analysis, and diagnostic thyroid lobectomy are added if FNAC is not diagnostic, which is likely in the case of FTC [36,37,38,39].

The definitive management of WDTC is either lobectomy or total thyroidectomy. Lymph node dissection, thyroid stimulating hormone (TSH) suppression, I-131 treatment, tyrosine kinase inhibitors, and external beam radiotherapy are added based on clinicopathological findings [1,8,40]. The risk level of the disease influences the choice of the management strategy for WDTC [8,41]. Well-differentiated thyroid cancers are heterogeneous tumors with divergent clinical behavior, response to treatment, and overall outcome [42,43]. Risk stratification of FTC includes the age of a patient, tumour size, evidence and extent of extra-thyroidal extension. The other factors that are important in the risk stratification of a patient who has FTC are the existence lymph node or systemic metastasis, pre-operative level of thyroglobulin (Tg) and completeness of surgical excision. The histological subtype, tumour differentiation of the tumour, immunohistochemistry, genomics, epigenomics, metabolomics and the changes in the micro-environment of the tumour also have an influence on the prognosis of WDTC and therefore guide of appropriate treatment [26,27,28,41,44,45,46,47,48,49,50,51]. Additional markers that have been found to be useful in the risk stratification of patients with TC include serum Vitamin D and the neutrophil-to-lymphocyte ratio (NLR) [52,53].

Although observation alone or lobectomy with lifelong follow-up may be appropriate for low-risk WDTC, patients whose tumors have a high risk of local recurrence or mortality should have total thyroidectomy with or without lymph node dissection, adjuvant or therapeutic I-131, aggressive TSH suppression, and intense monitoring during follow-up [1]. Although the 10-year survival of 90% of patients who are diagnosed with WDTC is over 95%, around 10% of the WDTCs are however unexpectedly aggressive and have a markedly reduced disease-free survival [2,3,8]. Patients who have been diagnosed with WDTC need a lifelong follow-up, which should be more intense in the first year following the initiation of treatment [1,54,55,56,57]. The follow-up program for patients who have WDTC includes clinical evaluation, neck ultrasound, monitoring of serum Tg, and radioisotope scanning, based on the patient’s risk level [1].

Thyroid cancer is a heterogeneous disease clinically and genotypically among patients and within itself and its metastases [58,59]. There is also high inter- and intra-observer variability during the interpretation of the results of imaging and FNAC or histopathological specimens of follicular-patterned neoplasms of the thyroid gland [60]. The current diagnostic and staging modalities used in WDTC are not able to accurately quantify the burden of the disease, and recurrence or progression of WDTC is sometimes detected late. Untreated WDTC has a propensity towards de-differentiating and becoming more aggressive as it progresses, and a previously low-risk and well-differentiated cancer may acquire new mutations, de-differentiate, and become aggressive and resistant to I-131 [8,9,61,62].

Follicular thyroid carcinoma cannot be diagnosed pre-operatively on FNAC because it can be confused with a follicular adenoma, rarely spread to lymph nodes, and has a different mutational landscape from that of PTC [9,25,47,63]. Additionally, FTC has a higher tendency, when compared with PTC and other thyroid malignancies, to present with systemic metastases from an occult primary tumor [57,64,65]. Additionally, patients who have FTC may present with hyperthyroidism [31,32,66]. The prognosis of patients with FTC is worse than that of classical PTC [56]. Table 1 contains a summary of the comparison of FTC with PTC.

The sections and subsections that follow discuss challenges during the diagnostic workup, management, and follow-up of patients who have FTC. The potential use of radiomics, pathomics, genomics, epigenomics, transcriptomics, proteomics, metabolomics, and lipodomics (multi-omics) to address some of the challenges and guide decision-making in the management of FTC is also discussed [58,73,74].

## 2. Diagnosis and Management of Follicular Carcinoma

Follicular carcinoma is the second most common malignant tumor of the thyroid gland and makes up around 10–15% of TCs [2,75]. There are two subtypes of FTC: minimally invasive follicular thyroid carcinoma (mi-FTC) and widely invasive follicular thyroid carcinoma (wi-FTC) carcinoma [9,11,35,62,76] (Figure 1).

The majority (86%) of FTC is diagnosed in women above the age of 40 years [5,77]. Less than 5% of FTC is heritable and related to syndromes such as FAP, Cowden’s disease, ataxia telangiectasia, and Li-Fraumeni syndrome [78,79]. Follicular carcinoma can develop from a pre-existing follicular adenoma (FA) [9,80,81]. Unlike with PTC, ionizing radiation and Hashimoto’s thyroiditis are not risk factors for follicular carcinoma [17]. The incidence of follicular carcinoma is decreasing globally [16,18,24,30,82,83]. Much of the decline in the prevalence of FTC is due to the iodine supplementation program, increasing expertise among pathologists, advances in immunohistochemistry, and the ability to perform mutational analysis [10,30,84]. However, around 72% of other types of thyroid cancer, including the follicular variant of PTC (FVPTC), are still mistakenly diagnosed as FTC [25,34,85,86,87,88,89]. Additionally, the OC was in the past erroneously considered a sub-type of FTC [90].

Follicular carcinoma of the thyroid usually presents as a nodular goiter, and rarely, patients who have FTC may present with hyperthyroidism or distant metastases from an occult primary tumor [57,64,91]. The common sites of distant metastases from FTC include bone, lung, and brain [57,92,93]. Lymph node metastasis occurs in less than 10% of FTC. The presence of lymph node metastases in a patient who has FTC should necessitate a review of the histopathology slides to rule out a missed FVPTC or other malignancies of the thyroid gland [94]. Because of its relative rarity, the diagnostic workup, staging, risk stratification, treatment, and follow-up of FTC are according to the guidelines developed for the management of PTC [1]. Follicular carcinoma is, however, clinically and genotypically different from PTC. Follicular carcinoma is more aggressive, has a greater propensity for haematogenous metastasis, and is associated with a shorter disease-free survival compared to PTC [28,85,94]. A longstanding FTC may de-differentiate and become a poorly differentiated (PDTC) or an undifferentiated/anaplastic thyroid carcinoma (ATC) [3,7,61,95,96,97,98,99]. Follicular thyroid carcinoma can also co-exist with other malignant tumors of the thyroid gland, which include medullary thyroid carcinoma (MTC) [100,101].

### 2.1. Diagnostic Work-Up of Suspected Follicular Carcinoma

Follicular carcinoma of the thyroid is rarely suspected pre-operatively except in patients who present with systemic metastasis [102,103]. The diagnostic work-up of a suspected FTC is like what is done for any patient presenting with a euthyroid nodular goiter and should include TFT, ultrasound, and FNAC [1]. None of the pre-operative investigations can confirm the diagnosis of FTC [89,102,103]. The majority of patients who have FTC are euthyroid, and the ultrasound examination is likely to show the same worrisome malignancy that is seen in PTC [104,105,106,107,108]. Among the worrisome features for malignancy on ultrasound of FTC is a solid hypoechoic lesion that is taller than wide and has increased intra-nodal vascularity, micro-calcifications, and an irregular border [104,105,106,107,109,110]. Follicular carcinoma is less likely to be multi-centric or multi-focal or be associated with regional lymph node involvement when compared with PTC [40,111]. Ultrasound also helps to pick up cervical lymph node metastasis and is complemented by a CT scan.

The Bethesda System of Reporting and Interpreting Thyroid FNAC (Bethesda System) is the most commonly used system to guide decision making regarding observation, repeat testing, additional pre-operative investigations, diagnostic lobectomy, or definitive surgical procedure for cancer [33,112,113]. The categories of the Bethesda system are non-diagnostic, benign, atypical, indeterminate, suspicious of malignancy, and malignant [33]. Result of FNAC of FTC is likely to yield a Bethesda III or IV lesion, which is an atypical or follicular neoplasm, respectively [33,114]. Follicular carcinoma is differentiated from FA by evidence of vascular and/or capsular invasion, which cannot be shown on FNAC, and FNAC is therefore not able to distinguish FTC from FA [89,102,115]. Follicular adenoma and FTC cannot be distinguished even after immunohistochemistry and mutational analysis [115]. The other thyroid conditions that may be mistaken for FTC following FNAC are adenomatous lesions of colloid goiter, FVPTC, OC, and the follicular variant of medullary carcinoma [89,116]. The diagnosis of FTC is only made after a lobectomy or total thyroidectomy. Diagnostic lobectomy is however unnecessary in up to 70% of the patients whose FNAC yielded a Bethesda System III or IV lesion, as the histology is likely to show a benign disease [33].

### 2.2. Staging of Follicular Carcinoma

The AJCC/TNM staging is used to stage WDTCs, including FTC, and parameters that are considered during the staging are the age of the patient, size of the tumor, evidence of extrathyroidal extension, presence of lymph node metastasis, and/or distant metastases [117]. Patients who are older than 55 years are at high risk of tumor progression or recurrence [2,118]. The categories that are used for tumor size are <2 cm, 2–4 cm, and >4 cm, whereas the nodal involvement is divided into involvement of the central nervous system, including level VII, or lateral cervical lymph nodes [117].

### 2.3. Risk Stratification and Associated Challenges in Follicular Carcinoma

The risk stratification of WDTCs considers gender and age of the patient, family history, type of tumor and histological subtype. The other parameters for risk stratification are tumor differentiation, degree of tumor necrosis, mitotic count, Ki67 index, mutational status and molecular subtypes. Lymph node status, thyroglobulin level, radio-iodine uptake, and PET/CT scan uptake are also important for risk assessment [26,35,51,94,119,120,121,122,123]. Follicular carcinoma is usually more aggressive than PTC, and the WI-FTC is likely to have distant metastases at presentation and lower disease-free survival [124,125]. The same risk scoring systems that are used in PTC are relied on for categorization of FTC, and among them are TNM staging; metastasis, age, completeness of resection, invasion, and size (MACIS); age, gender, extra-thyroidal extension, and size (AGES) and age, metastasis, extra-thyroidal extension and size (AMES) [119,121,125]. Low-risk FTC is a tumor that is less than 4 cm in maximum diameter and has no extra-thyroidal extension, lymph node, or distant metastasis, whereas an intermediate- or high-risk tumor is larger than 4 cm. A patient who has an FTC that is not completely excised, has extensive extrathyroidal extension, or has metastasized to lymph nodes or distant organs has a high-risk tumor [41,126]. Markers that have been used for risk-stratification of WDTC, including FTC, include vitamin D level and neutrophil-lymphocyte ratio (NLR) [52,53].

Patients who have FTC and are older than 55 years old are at increased risk of tumor recurrence or metastasis [2,118]. The organ involved influence the prognosis of patients who have systemic metastases. Patients who have pulmonary metastases from FTC generally do better than those who have metastases to bone, brain, liver, and other organs [93,127]. The prognosis depends on the volume of metastasis in an organ [93,127]. All the risk scoring systems, including the AJCC/TNM systems, are however not able to accurately quantify the burden of the disease, especially in patients with FTC [128]. The pathological diagnosis of FTC and its subtypes is dependent on the availability of expertise to perform the histopathological analysis [87]. Like other cancers, FTC is heterogenous, the aggressive component of the tumor may be overlooked during histopathological assessment, and a patient who has supposedly had mi-FTC and has had an appropriate lobectomy or total thyroidectomy may present years later with distant metastases [45]. The metastases are likely to have been there and were missed during the pre-operative assessment.

### 2.4. Management of Follicular Carcinoma and Related Challenges

The management of FTC follows that of PTC, and the selection of the package of care depends on the risk of recurrence and mortality [13,111]. Lobectomy or total thyroidectomy is the primary curative management of FTC. The other treatment strategies of FTC are added depending on the level of risk [51,56,59,126,129]. Lymph node metastasis occurs in less than 10% of FTCs, and lymphadenectomy is only performed if the involvement is confirmed clinically and/or following imaging investigation [40,111]. Additional treatment that may be required during the treatment of FTC includes radioactive iodine, TSH suppression, metastectomy, tyrosine kinase inhibitors (TKIs), multi-kinase inhibitors (MKIs), and de-differentiation therapy [8,9,61,62,130]. Evidence of de-differentiation of the tumor, which is likely if the tumor is locally advanced with extensive extra-thyroidal extension or is metastatic, may also influence the decision regarding further management of FTC [3,8,61,98,131,132,133].

#### 2.4.1. Lobectomy versus Total Thyroidectomy in FTC

The diagnosis of FTC is not possible on pre-operative FNAC and is usually made following a diagnostic lobectomy, and the need for a complete thyroidectomy is only considered thereafter based on the size of the tumor and whether a patient is at high risk for recurrence or metastasis [126]. Lobectomy alone is appropriate for a tumor that is 1–4 cm in maximum diameter without high-risk features based on clinical, histological, immunohistochemical, or mutational analysis [56,59]. Total thyroidectomy should be the standard of care for FTC if the primary lesion is more than 4 cm in diameter. Mi-FCT usually has a benign course and does not require a complete thyroidectomy or radioactive iodine ablation. However, areas of major invasion may be missed during histopathological analysis, leading to the erroneous labeling of a wi-FTC as a mi-FTC [11]. The ability to perform mutational analysis is also not universally available.

Another parameter to consider is multifocality. Although the FTC is less likely to be multifocal when compared with the PTC, lobectomy alone in patients who have multifocal disease may lead to recurrence of the tumor [15,68,87]. The tumor that was left in the other lobe may de-differentiate and become a PDTC or ATC [8,54,72,124,131]. The extent of the primary tumor is another predictor of a poorer outcome in patients with FTC. Patients with macroscopic extra-thyroidal spread are at high risk of local recurrence or metastasis and should therefore be offered total thyroidectomy [68,134]. A total thyroidectomy is mandatory if the FTC is metastatic, as post-operative radioactive iodine would be required.

#### 2.4.2. Radioactive Iodine Therapy and Dosimetry in the Management of FTC

Radioactive iodine is indicated in the management of intermediate- and high-risk FTC for the ablation of a remnant or as adjuvant therapy or treatment of metastatic disease following total thyroidectomy [13]. The aim of ablation therapy is to prevent local recurrence and to facilitate the use of Tg as a tumor marker for monitoring during follow-up [2,3,13,135]. The standard dose of I-131 for ablation of the remnant of the thyroid is 30 mCi. The dose for ablation of the remnant is administered around 6 weeks after a near-total or total thyroidectomy, when the s-TSH level is expected to have risen to 30 mIU/mL or higher [136]. A high level of s-TSH is achieved after a period of deferral of thyroxine replacement or by using recombinant TSH [72,135]. Adjuvant I-131 is to target presumed micrometastases from an intermediate- or high-risk FTC, which were however not picked up during pre-operative evaluation [56]. A dose of up to 150 mCi is necessary for ablation of metastases from FTC and can be increased to around 200 mCi if the metastases are extensive [3,28,135,136].

The dose of I-131 is adjusted considering the age of the patient, co-morbidities, the organ involved, and the burden of the metastases [72,135]. Clinical dosimetry is, however, less accurate when compared with radioisotope-based dosimetry. Dosimetry using I-124, I-123, or I-131 is useful for guiding the most effective dose of I-131 while reducing the likelihood of side effects [13]. Radioisotope-based dosimetry may also assist in the early identification of dedifferentiated metastases from FTC that are not trapping the iodine to avoid futile treatment. Problems associated with the use of I-131 during the management of FTC include side-effects like bone marrow suppression, xerostomia, infertility, severe hypothyroidism during the suspension of thyroxine replacement, and the development of a second primary malignant tumor [127,134,137,138]. The additional concerns of the use of I-131 for treatment in patients who have extensive lungs or brain metastases are that they may develop pulmonary fibrosis that may lead to respiratory failure or brain oedema, respectively [92,127,138]. Other challenges linked to the management of FTC using I-131 include the heterogeneity of cancer. The FTC may not be I-131 avid and therefore resistant to radioactive treatment. Resistance to I-131 is likely in patients who have metastatic disease [3,47,95,99,132,135,136,139,140,141,142]. Around 10% of TC cancers are under-staged and deemed low-risk, not needing I-131 ablation or adjuvant therapy). Similarly, areas of WI-FTC may be missed during histopathological analysis [11,86].

#### 2.4.3. Thyroid Stimulating Hormone Suppression during Management of FTC

Patients who have had a total thyroidectomy need thyroxine for replacement and suppression. All patients who have FTC require TSH suppression regardless of the risk level [13,117,126]. The intensity of TSH suppression depends on the age of the patient, the presence of co-morbidity, and the level of risk for tumor recurrence or mortality [126,143]. The TSH suppression may be severe, moderate, or minimal [144]. Severe TSH suppression is when the TSH level is below 0.01 mU/L, moderate suppression is 0.01–0.1 mU/L, and mild suppression is 0.1–0.5 mIU/L [144]. Severe and moderate TSH suppression increase the risk of cardiac and musculoskeletal side effects. Cardiac side effects of high doses of thyroxine include atrial fibrillation and an increased risk of ischemic heart disease [145,146]. Osteoporosis is among the most severe musculoskeletal complications of a suppressive dose of thyroxine [147]. Additional side effects of extreme TSH suppression include depression and weight loss [147]. Furthermore, some of the patients may be misclassified as having low-risk instead of high-risk FTC and be erroneously placed on a less intense TSH suppression program [148]. The other problem with TSH suppression in the management of FTC is in tumors that have acquired aggressive mutations, have de-differentiated, and are no longer responsive to treatment [9,83,142].

#### 2.4.4. Management of FTC with Extensive Extra-Thyroidal Extension

Some of the patients who have FTC may present with a tumor that has invaded the aerodigestive tract [49,149]. Mortality in half of the patients who have TC is related to local invasion of the upper aerodigestive tract and major vessels in the neck by the tumor [150]. Despite the extensive local invasion, curative resection may still be feasible unless the tumor is invading the carotid vessels or prevertebral fascia [149,151]. The nature of surgical resection may be extensive en-block resection of the larynx or oesophagus or a shaved excision [149]. Other treatment options for FTC with extensive extra-thyroidal extension include external beam radiotherapy and radiofrequency ablation [96,152]. Extensive extra-thyroidal extension is a marker of a high-risk FTC that is likely in tumors that have de-differentiated and become PDTC or ATC [96,149]. The area of de-differentiation might have been missed during the sampling and analysis of the FNAC or post-thyroidectomy specimen [96]. In some cases, the cancer might have already spread systemically, making the sometimes-debilitating en-block resection a futile operation.

### 2.5. Management of Metastatic Follicular Carcinoma and Related Challenges

The incidence of systemic metastases in FTC has been reported at 5–23% [150,153]. Follicular cancer rather than papillary thyroid cancer is more likely to metastasize to distant organs. Common sites of metastases for WDTC are the lungs, bones, brain, skin, and liver [65,93]. Male sex, older age, tumor size greater than or equal to 4 cm, vascular invasion, and lymph node involvement are some of the risk factors for systemic metastasis from FTC [153]. Systemic metastases are likely in TC with evidence of extra-thyroidal extension [153]. Poor prognostic features in patients who have metastatic FTC include age over 55 years, size of metastasis above 10mm in the maximum diameter at detection, and a high neutrophil to lymphocyte ratio). Well-differentiated FTC, especially in children, and FVPTC are relatively more I-131 avid and are therefore likely to respond to ablation or treatment [123].

The choice of treatment for FTC considers the patient’s age, comorbid conditions, sites, and number of metastases [1,5,40,51,63,72]. The prognosis of metastatic FTC is better in children and young adults, patients who have isolated lung metastases, metastases less than 5 cm in maximum diameter, and oligometastatic disease [1,2,26,40,51,56,63,66,71,72]. Patients who have intermediate- or high-risk FTC are likely to have systemic metastasis and be given adjuvant I-131 [1]. Radioactive iodine treatment is for the management of patients whose metastases are overt, and a dose of up to 200 mCi is used [117]. Treatment with I-131 is added after a patient has had a total thyroidectomy with or without resection of metastases that are amenable to resection [72]. Some of the problems associated with I-131 treatment of metastatic disease include the side effects, which are sometimes severe, like bone marrow suppression, infertility, and the development of another cancer [72,127]. A higher dose of I-131 is to be avoided in patients who have extensive pulmonary metastases and borderline lung function, multiple brain metastases, or large metastases to the vertebra, as it may lead to complications like pulmonary fibrosis, brain oedema, or instability of the spine [117,127]. However, extensive metastases are less likely to be iodine-avid as they would most probably have mutated and de-differentiated.

Patients who have metastatic FTC are in the high-risk group for WDTC and should be placed on an intense or high TSH-suppression protocol, unless there are contraindications, or it is not tolerated. The other options in the management of metastatic FTC are surgical excision, external beam radiotherapy, local ablative therapy, and kinase inhibitors. Surgical excision is preferred for the treatment of isolated metastases from FTC to most organs, including the lung, bone, brain, and liver. The TKIs are useful alone or when combined with other multi-kinase inhibitors (MKIs), especially if the FTC has dedifferentiated and is no longer retaining I-131 [72]. Examples of TKIs include lapatinib and vemurafenib [154]. The MKIs that are useful against WDTC include cabozantinib, lenvatinib, and sorafenib [71,155,156].

#### 2.5.1. Pulmonary Metastases

The lung is the commonest site of distant metastases from FTC. Patients who have pulmonary metastases from FTC have a reduced 5-year and 10-year survival of 68.5% and 54%, respectively [153]. Lung metastases in FTC are likely to be macronodular when compared with secondary tumors in PTC [153]. The success rate of I-131 in the management of pulmonary metastases from FTC is lower than the 58% that is achieved in cases of metastatic PTC [153]. Patients who have FTC are usually older than those who have PTC and are therefore more likely to have co-morbidities [68]. A tolerable dose of I-131 depends on the baseline lung function and the extent of the disease, as ablation of the extensive pulmonary metastasis and their replacement with fibrous tissue may push the patient to irreversible respiratory failure. The other options for the treatment of pulmonary metastasis from FTC include surgical excision, percutaneous ablation, stereotactic radiotherapy, and tyrosine kinase or multi-kinase inhibitors [153].

The 18F-FDG PET/CT is useful for the investigation of recurrent or metastatic FTC when it is no longer I-131 iodine-avid. De-differentiation of FTC is associated with an increase in the expression of glucose transporter receptor type 1 and therefore high uptake of 18F-FDG PET/CT [72]. The uptake of DOTA PET by some of the metastases from FTC that have lost the ability to trap iodine is even higher than that of 18F-FDGvPET/CT, which raises the possibility of utilizing 177Lu as a potential theranostic agent [153]. There are numerous society guidelines for the management of metastatic disease from differentiated thyroid cancers, and most of them address WDTCs as a group and not specifically FTC [45]. The guidelines are mainly for PTC. Ongoing research for markers predictive of poor prognosis includes the investigation of Telomerase Reverse Transcriptase (TERT) promoters and RNA H-19. Challenges associated with the management of metastatic FTC include delay in their detection, underestimation of the extent of the disease, and dealing with metastases that do not take I-131 [150]. Sometimes, metastases are missed, and FTC is erroneously labeled as low-risk and treated without total thyroidectomy, I-131 ablation, or adjuvant therapy and TSH suppression, only to present years later with systemic metastases.

#### 2.5.2. Bone Metastases

Patients with FTC are more likely to develop bony metastases than those who have PTC. The bone is the second most common site of metastases from FTC. The prognosis of patients who have bone metastases is poorer when it is compared with that of patients in whom the FTC has spread to the lungs. Additionally, bone metastasis from FTC is less sensitive to I-131. The bone metastases may be symptomatic or detected on CT, MRI, 18 FDG PET CT, SPECT/CT, or WBS using I-123, I-124, or I-131. Patients who have metastatic FTC are treated with I-131 together with TSH suppression unless the metastases are no longer iodine-avid, but FTC tends to acquire aggressive mutations as it progresses and metastases. Radioisotope-based dosimetry, although not uniformly practiced, can provide guidance on the safety and efficacy of I-131 treatment and help avoid unnecessary toxicity and treatments in patients in whom the metastases are not responding to the treatment [153]. Usually, close to 30% of patients who have metastatic FTC do not respond to I-131 [153].

#### 2.5.3. Brain Metastasis

Brain metastases from FTC are rare and occur in less than 1% of the patients, and over 70% of the patients who have brain metastases from FTC are likely to have synchronous pulmonary metastases [28,157]. The prognosis of patients who have FTC with metastasis to the brain is poor, and the overall survival is less than 3 years even with treatment [158]. Resection for an isolated brain metastasis from FTC is preferred if it can be excised [157,159]. Severe TSH suppression should be part of the treatment of brain metastases from FTC. Treatment with I-131ne is effective if the metastases are iodine-avid and not extensive. The other options for the treatment of brain metastases from FTC include stereotactic radiosurgery, external beam radiotherapy, and multi-kinase inhibitors [28,157].

### 2.6. Management of De-Differentiated FTC

De-differentiation of FTC is a process that happens during tumor progression, and a de-differentiated cancer is likely to have new mutations and become aggressive and resistant to conventional therapy, such as I-131. De-differentiated FTC and PTC, PDTC, and ATC account for most of the deaths due to TC. The more prevalent mutations in de-differentiated TCs, including FTC, are *BRAF*, *RAS*, *TP53*, *TERT* promoter, and *P13K/AKT/mTOR* pathway effectors [98,160]. Both SPEC/CT and PET/CT are useful for evaluating the burden of metastases in cases where the tumor is no longer iodine-avid. The treatment of de-differentiated FTC is like that of PDTC and ATC and includes the use of TKIs or MTIs. Some of the de-differentiated FTCs may overexpress somatostatin receptor type 2 (SSTR2) and could be evaluated with 68Ga-DOTANOC PET/CT. Patients whose metastases demonstrate high uptake of 68Ga-DOTANOC may respond to treatment with 177Lu-Dotatate [161].

### 2.7. Follow-Up of Patients Who Have Follicular Carcinoma and Associated Challenges

Guidelines relating to the follow-up of FTC are often inferred from those of PTC, even though FTC forms a unique subgroup of DTC (differentiated thyroid cancer) and, therefore, requires specific considerations. A multidisciplinary approach with well-established communication channels between clinical endocrinologists and surgeons, pathologists, oncologists, radiologists, and nuclear physicians should form the backbone of the care of a patient who has FTC. Placing the patient in the center goes a long way towards improving the quality of care and eventually securing a good outcome. An important tool in achieving this is risk stratification, which consists of both. Patients who have been treated for FTC require lifelong follow-up, regardless of the initial risk level of the tumor [121]. The follow-up is intense in the first year following definitive treatment. Dynamic risk stratification, clinical evaluation, serum basal and stimulated Tg levels, and neck ultrasound are used during follow-up monitoring. Concurrent testing for the existence of Tg antibodies must be done to reduce the likelihood of falsely low levels of Tg.

The success of the management of the FTC should be determined within the first 3–6 months. Patients can be categorized as possibly cured both biochemically and anatomically, biochemically incomplete, or anatomically residual. The primary purpose of the stringent follow-up of cases of FTC is for the timeous diagnosis of persistence or local recurrence of the tumor or the appearance of new metastases. It is also critical that the de-differentiation of FTC is detected and managed early, using agents such as TKI or de-differentiation therapy to improve the chance of long-term survival [8,72,131]. The follow-up of patients on TKIs or MKIs must be regular and stringent to watch for side-effects. The most dreadful complication is bleeding, which, although rare, may be fatal [70].

It is not usual for metastases from FTC to lie dormant and unsuspected, even for tumors that are deemed low-risk clinically, pathologically, and radiologically [54,57,148,162]. Follicular thyroid carcinoma may acquire aggressive mutations as it progresses. Some of the aggressive mutations that the FTC can acquire include TP53, TERT, and PI3K/AKT/mTOR mutations [3,6,9,47,61,98,99,132,163,164].

### 2.8. Outcome of Treatment of Follicular Carcinoma

The outcome of patients who have FTC depends on the stage of the disease at presentation. The available staging systems for WDTC are not able to accurately predict disease-specific mortality [35]. The AJCC is the most commonly used staging system. Unfortunately, the AJCC/TNM staging on its own is not enough to direct and monitor a specific therapy. Other risk stratification systems are considered to improve the care of patients with DTC. The inclusion of intra-operative findings, histopathological changes, and genomics improves the prognostication of FTC.

## 3. Recent Development in the Genomics of Follicular Carcinoma and Other Thyroid Tumors

The development of cancer, including its progression and metastasis, is usually driven by genetic aberrations due to mutations and re-arrangements in genes that regulate pathways that are controlled by trans-membrane mitogen-driven activating kinases [9,99]. The main pathways in tumorigenesis in WDTCs are the mitogen-activating protein kinases, phosphatidylinositol 3, and *Wnt*/mammalian target of rapamycin (*Wnt/mTOR*) pathways [160]. The mutations that drive the protein kinases may be oncogenes or tumor suppressor genes [8,9]. The effects of the mutations or re-arrangements on the pathways may include the initiation of the tumor, proliferation, failure of apoptosis, neovascularization, neo-lymphogenesis, neurogenesis, epithelial to mesenchymal differentiation, and acquisition of metastatic potential [9]. The genetic aberrations are usually mutually exclusive but may rarely occur in combination [47,83]. Changes in the manifestations of a gene may also occur with or without mutation and be driven by epigenetic alterations.

The genes that are commonly dysregulated in TC are *BRAF*, *RAS*, *PTEN*, *TERT* and *TP53* [9,58,61,71,132,165]. Some of the receptors that are controlled by genes that are either up- or down-regulated in thyroid cancers are located within the cell membrane, whereas others are found inside the cells. An example of transmembrane receptors are the various types of tyrosine kinase receptors (TKRs) such as *RET*, *VEGFR*, *EGFR*, *PDGFR*, *MET* and *c-KIT* [71]. Mutations of the *RAS*-related genes (*K-RAS*, *N-RAS* and *H-RAS*) usually occur in the earlier phases of tumorigenesis in WDTCs [9]. The RAS mutations are more prevalent in FTC but have also been reported in FA, FVPTC, and OC [47,166]. Epigenetic changes, including those caused by non-coding RNAs, may modify the impact of the genetic changes [115,167]. The genotype and behavior of WDTCs, including FTCs, change as the tumor grows. For example, the sodium iodide symporter (NIS) stops functioning as FTC progresses, which makes the tumor less avid and responsive to I-131 treatment [136]. The number of mutations tends to increase as FTC progresses and de-differentiates [98,133,168]. The dysregulation of the *p53* and *Wnt* pathways occurs later during the progression of TC. Figure 2 is a schematic illustration of the common mutations that are implicated in tumorigenesis, progression, and de-differentiation of FTC compared to PTC.

## 4. Multi-Omics of Follicular Carcinoma and Other Thyroid Tumors

A vast amount of information is collected during the clinical evaluation, diagnostic work-up, staging, and follow-up of patients with FTC. The collected information may be categorized into biographic, life-style, environmental, clinical, radiological, biochemical, cytopathological, histopathological, or genetic categories [169]. The big data that is obtained during whole genome sequencing adds to the enormity of the collected information [170,171,172,173,174,175]. Currently, a minute fraction of the information that is generated during the investigation is used to guide decision-making during the care of patients who have cancer, including FTC. Even where the data is used, it tends to be fragmented, and the analysis rarely incorporates results from all the omics. Recent advances in computing have allowed for the mining of the data as part of the different types of omics that include radiomics, pathomics, and genomics [72,73,74,176,177,178,179,180]. The field of multi-omics is a process of integrating the personal background of a patient and results from clinical evaluation, radiomics, pathomics, genomics, proteomics, and metabolomics to guide decision-making. Multi-omics is the consolidation and integration of findings from several omics to address some of the challenges that are encountered during the diagnostic work-up, staging, risk-stratification, management, and follow-up of most cancers, including FTC.

### 4.1. Pathomics in the Diagnostic Work-Up of Follicular Carcinoma

Several investigations have been trialed to distinguish FA from FTC, but none can confirm FTC before lobectomy or thyroidectomy. Another challenge in the assessment of the pre-operative investigation of FTC is its heterogeneity [139,178]. A site in the tumor that contains evidence of invasion or nuclear features of PTC may be missed during sampling, leading to the mis-classification of the tumor [34,35,86,88]. Among the additional investigations that are used during the attempts to rule in or rule out cancer in a thyroid nodule with an indeterminate FNAC are elastography and elastometry, an I-123 scan, a CT scan, and an MRI [181]. Thyroid neoplasms are associated with either an up- or down-regulation of the expression of proteins that regulate proliferation, differentiation, mobility, and apoptosis of cells that may be assessed through immunohistochemistry [9,136]. The application of AI can help distinguish FA from FTC [182,183]. Digital slides may be generated, which may enhance the thoroughness of the evaluation of FNAC or histopathology slides.

### 4.2. Genomics of Follicular Carcinoma and Other Thyroid Malignancies

The most frequently encountered mutation in PTC is the *BRAF V600E* mutation, whereas the RAS mutation is more common in FTC [58,97,99]. The types of mutations seen in FTC often do not differ from those seen in FA. The diagnosis of FTC can only be made following a lobectomy [58,69]. The other genes that are mutated or re-arranged during tumorigenesis and progression of WDTC are the v-raf murine sarcoma viral oncogene homolog B1 (*BRAF*), rat sarcoma virus (*RAS*) and mitogen activated protein kinase (*MEK*). Mutations in patients who have WDTC may also involve the extracellular signal-regulated kinase (ERK), tumour protein 53 (TP53), telomerase reverse transcriptase (*TERT*), Phosphatase and TENsin homolog deleted on chromosome 10 (*PTEN*) and Retinoblastoma-1 (*RB1*) genes [7,9,14,35,47,86,87,88,89,90,91,92,93,94,95,96,97,98,98,99,100,101,102,103,104,105,106,107,108,109,110,111,112,113,114,115,116,117,117,118,119,120,121,122,123,124,125,126,127,128,129,130,131,131,132,133,134,135,136,137,138,139,140,141,141,142,143,144,145,146,147,148,149,150,151,152,153,154,155,156,157,158,159,160,161,162,163,164,165,166,166,167,168,169,170,171,172,173,174,175,176,177,178,179,180,181,182,183,184] (Table 2).

Thyroid cancers that harbor *BRAF V600E* are likely to be more aggressive and less I-131-avid [98,99]. The *RAS* gene associated mutations (*Kirsten-RAS*, *Neuroblastoma-RAS*, and *Harvey-RAS*) have not been conclusively linked to the clinical behavior of FTC or other thyroid malignancies and are not useful for differentiating FA from FTC [80,164,195]. The TP53 mutation happens in the later stage of the development of FTC and is likely to be associated with the de-differentiation of the cancer to a poorly differentiated or anaplastic tumor [95,98,99,140]. The mutations, together with the epigenetic and transcriptomic changes, are sometimes used to risk-stratify WDTCs and predict the likelihood of local recurrence, micro-metastasis, and disease-free survival. The non-coding RNAs that may be under- or over-expressed and can be used to predict the clinical behavior of FTC [2,45,50,167,189]. A change in the level of histone acetylation or methylation can sometimes also influence the development, progression, and response of FTC to treatment [189].

### 4.3. Epigenomics of Follicular Carcinoma and Other Thyroid Tumors

Tumorigenesis or tumor progression not always linked to the down- or up-regulation of a gene but to epigenetic changes. The epigenetic changes may be due to modifications of histones or post-transcriptional changes. Histone forms a complex with the chromosome. Acetylation, methylation, phosphorylation, and other forms of modification of histones may lead to the down- or up-regulation of a gene [196,197]. Non-coding RNAs may modify the activities of messenger RNA (m-RNA) or indirectly through their effect on ribose binding proteins (RBPs). The types of non-coding RNA include circular RNA (crRNA), long non-coding RNA (ln-RNAs), and micro-RNAs (mi-RNAs). The up- or down-regulation of non-coding RNAs is associated with the development and progression of FTC and other thyroid malignancies.

#### 4.3.1. Histone Modification

The epigenetic modifications reported commonly in patients who have FTC and other thyroid malignancies involve acetylation or methylation of histones. The pattern of acetylation or methylation can be useful to resolve a diagnostic dilemma or guide treatment. Histone modification may induce cancer by either augmenting the oncogenes, interfering with the activities of tumor suppressor genes, or inhibiting the apoptosis of cells [197,198].

#### 4.3.2. Non-Coding RNAs

Most of the genes (75% of them) that are transcribed into RNA do not translate into proteins but result in the production of non-coding RNAs (Liu et al., 2021). Among the non-coding RNAs are micro-RNA (mi-RNA), long non-coding RNA (ln-RNA), circular RNA (circRNA), and PIWI-interacting (piRNA) (Yan and Bu, 2021). Non-coding RNAs have been shown to play a crucial role in the initiation, proliferation, and progression of many cancers, including TCs [45,50]. Non-coding RNAs are variably expressed in TCs, including FTC, and may be used during diagnostic investigation, risk stratification, or to guide treatment [45,199,200]. Examples of mi-RNA and ln-RNA that are useful during the diagnostic investigation or treatment of FTC is miR-146b and H19 [45,199].

### 4.4. Proteomics of Follicular Carcinoma and Other Thyroid Malignancies

Proteins are very intimately involved in the regulation of the behavior of cancers, including FTC. Some of the mutations, fusions, or re-arrangements of genes that occur during tumorigenesis lead to either an increase or a decrease in the synthesis of proteins and their metabolic products. The level of certain proteins and their metabolites is either increased or decreased, and their expression profile may be used for the diagnosis or risk stratification of cancers. There is a difference in the profile of proteins that are increased in the serum of patients who have thyroid nodules that may be used to assist in differentiating FTC from FA or PTC [201]. Immunohistochemistry on the FNAC specimen may be used in the diagnostic evaluation of indeterminate thyroid nodules [183]. Among the markers that are used during risk stratification of WDTCs, including FTC, are galectin-3, HMBP-1, E-cadherin, PGER1, PD-L1, and TFF-1 [8,39,44,49,96,110,183,200,202,203]. Table 3 contains a list of some of the tyrosine kinase receptors whose expression is altered during the tumorigenesis and progression of FTC and other TCs.

### 4.5. Metabolomics of Follicular Carcinoma and Other Thyroid Tumors

The majority of cancers are associated with changes in the metabolic processes within the cancer and the TME [200]. Some of these changes have a significant influence on the behavior of tumors, including their local aggression and their ability to spread to distant sites. Cancer cells are characterized by rapid division and multiplication, which are energy-dependent processes [209]. Cancer cells can alter their metabolic process and predominantly utilize the pyruvate acid cycle to obtain their energy regardless of oxygen status, the so-called Warburg effect [209,210]. Adipocytes in the TME can promote tumor growth, progression, and metastasis through the secretion of adipokines). The differentiated profile of metabolites in the tumor itself, TME, and fluid in a patient with a thyroid nodule whose FNAC result is inconclusive may be used to differentiate FTC from FA or other follicular-patterned lesions [201]. The study of the change in levels of these molecules (metabolomics) is assessed using nuclear magnetic resonance spectrometry or mass spectrometry. The other strategy to investigate the profile of metabolites in tissues or fluids from patients who have FTC or other cancers is gas or liquid chromatography [209]. Metabolomics may also be applied to FNAC specimens from thyroid nodules to help distinguish benign from malignant causes. Among the metabolites that expressed differentially in patients who have benign and malignant conditions of the thyroid gland are lactate, glutamate, methionine, hypoxanthine, phenylalanine, taurine, and tyrosine [209].

### 4.6. Liquid Biopsy in Follicular Carcinoma and Other Thyroid Tumors

Differentiating FTC from FA and other follicular neoplasms remains a challenge despite the increasing availability of expertise in pathological analysis, immunohistochemistry, and mutational analysis. Because of the heterogeneity of FTC, the specimen for FNAC, which was taken from a small area, may not be a true reflection of the type and subtype of TC [139,179]. Confirmation of FTC is challenging even after thyroid lobectomy, as areas where there is capsular or vascular invasion may be missed. Another of the challenges that is persistent is the inability to accurately diagnose the extent of vascular invasion, which is required for the differentiation of mi-FTC from wi-FTC [86,211]. That a low-risk WDTC has become high-risk or de-differentiated may also go unnoticed following FNAC, diagnostic lobectomy, or total thyroidectomy [10,54,55,57,86,98,162]. Tumor recurrence and metastasis from FTC are often diagnosed late, when they are extensive and have acquired aggressive mutations [54,96,148]. Liquid biopsy provides a non- or minimally-invasive approach to the diagnostic work-up, risk stratification, and monitoring of patients who are suspected of having or are on treatment for FTC [212]. Liquid biopsies sample blood, saliva, or urine to look for circulating tumor cells, cell-free DNA, or extracellular vesicles (exosomes) that are released from the tumor [199,213,214]. Liquid biopsy is useful for differentiating FA from FTC, assessing the TME, early identification of metastases, and a de-differentiated FTC [163,199,212,215,216]. The existence of mutations such as the BRAF600E may be confirmed through a liquid biopsy done on the serum of a patient who has WDTC [199].

### 4.7. Radiomics and FTC

Facilities and expertise for liquid biopsy are not freely available. Ultrasound of the thyroid is operator-dependent, and the level of expertise influences the accuracy of the interpretation of FNAC, imaging, histopathology, and genomic studies. There is a possibility that the FNAC or post-lobectomy may either miss a cancer or underestimate the histological grade and aggressiveness of the tumor as the entirety of the tumor and the tumor micro-environment are not assessed [102,162,179,206]. Artificial intelligence is useful in the diagnostic work-up of a thyroid nodule, especially when the FNAC result is indeterminate [73,177,178,179]. An integration and concurrent analysis of findings from ultrasound and elastography, radioisotope scan and MRI is most likely going to increase the accuracy of predicting the likelihood of FTC. The extent of vascular or capsular invasion of the FTC into may also be determined without the need for an invasive biopsy or a costly and potentially risky diagnostic lobectomy [176,217,218,219,220,221,222,223] Radiomics enables segmentation of the tumor and facilitates a thorough analysis of the tumor and the TME [163,224,225].A thorough analysis of a thyroid nodule and the TME may enable a “virtual” biopsy of the lesion and accurate histological diagnosis and grading of the tumor, like what would be achieved following a tissue or liquid biopsy. Some of the imaging investigations, such as [^18^F] FDG PET/CT and [^18^F] FDG/MRI, can localize metastases from FTC when there is discordance between the serum Tg level and the results of a whole-body radioiodine scan (WBS) [72,131,142,226].

## 5. Artificial Intelligence and Management of FTC

Follicular carcinoma is a heterogeneous disease. Its heterogeneity is manifested in its clinical presentation, findings on diagnostic investigation, response to treatment, and long-term outcome [35,150]. A vast amount of information is collected during history taking, physical examination, diagnostic and staging investigations, and monitoring following definitive management [170,173]. Analysis of the findings is influenced by the level of expertise, which explains the often very high intra- and intra-observer variation regarding histological confirmation of the type and subtypes of TC, differentiation, and changes in the TME [227,228]. Consequently, it is not unusual for a high-risk TC to be under-staged and undertreated until it recurs or metastasis becomes evident [54,150]. The inability to incorporate all the relevant information that is collected during the evaluation of patients is evident in those who have FTC and are still not cured, even where management decisions are made through a multidisciplinary team. The use of artificial intelligence (AI) in healthcare has led to an improvement in the accuracy of diagnosis and decision-making during the management of benign and malignant disease [179,229].

Artificial intelligence relies on advanced computing to make decisions based on the information obtained from multiple sources [230,231,232]. Artificial neural network (ANN) and convolutional neural network (CNN) are examples of AI, which vary based on the number of inputs and hidden layers that are involved before a decision is reached [183,229,233] (Figure 3).

Using CNN would be ideal to deal with problems like indeterminate FNAC result. Other problems that can be addressed if AI is implemented is the incorrect risk-stratification of patients who have FTC and the appropriate level of TSH suppression in intermediate and high-risk patients. Additional benefits of AI is accurate calculation of I-131 dose for ablation, adjuvant therapy or treatment, a delay in the detection of progression, recurrence, and de-differentiation of the tumor, and prediction of the response to treatment [183,221,223,231,234]. The use of AI in the management of FTC can simultaneously incorporate background personal, life-style, and environmental factors (digital personal twin), imaging results (radiomics), FNAC and histological results (pathomics), mutational analysis (genomics), histone modification and transcriptomics (epigenomics), and post-translational changes (proteomics and metabolomics).

The complexity of FTC starts with diagnostic investigations, continues through staging and decision-making regarding appropriate management, and ends with how to timeously detect tumor recurrence or metastasis. Although the majority of patients with FTC are clinically and biochemically euthyroid, some may be thyrotoxic [31]. Follicular carcinoma does not share the same ultrasound features of malignancy that are seen in PTC and other TCs [235]. Furthermore, the clinical presentation and ultrasound findings in FTC are usually like those in FA. Fine needle aspiration cytology cannot distinguish FTC, FA, FVPTC, and other follicular-patterned lesions [215]. The application of AI can predict the diagnosis of cancer in a thyroid nodule that returned a Bethesda III or IV following FNAC [223,236]. The performance of AI-based technology sometimes surpasses that of less experienced radiologists, which would be useful in areas where resources are limited [236,237,238].

Artificial intelligence programs are also able to predict the mutational status of FTC and other malignancies of the thyroid gland [69,155,239]. Additionally, based on ultrasound features, which include the diameter of FTC, the use of AI can accurately predict the existence of systemic metastases and lead to timeous escalation of treatment [235]. Artificial intelligence is also useful for the identification of high-risk MI-FTC that have metastatic potential, ensuring the early initiation of aggressive treatment [45]. The other benefit of using AI is in the analysis of cytological specimens [240]. Machine learning applications can accurately predict FTC when the FNAC result is inconclusive [240,241]. The application of AI-aided evaluation of FNAC specimens may also determine the nature of TME and predict the likelihood of the existence of mutations such as the BRAF600E mutation [239,242]. Imaging techniques such as plain x-ray and CT scan may miss small lung metastases from FTC and lead to erroneous categorization of a patient as being at low risk of persistence or recurrence of the tumor. Using a combination of demographic, clinical, imaging, and histopathological features, AI can accurately predict the existence of pulmonary metastases.

Multiple omics data is collected during the investigation, treatment, and follow-up of a patient who has FTC [243]. Although the information is presented at an MDT, the sets of data that are generated are vast and are often interpreted in isolation. Making decisions based on one set of omic data may lead to over- or under-treatment of FTC, like other cancers [244]. The availability of AI provides a platform for the integration and analysis of the data obtained from various omics to enable computer-aided diagnosis, risk stratification, choice of treatment, and follow-up instead of the current treatment guidelines, which oversimplify the complexity and heterogeneity of FTC [245] (Figure 4).

## 6. Conclusions

The challenges associated with the management of FTC cover the entire continuum from diagnosis, risk stratification, treatment decision, and follow-up. Whilst most FTCs are mi-FTC, some of the wi-FTC may be missed, leading to an underestimation of the risk of the tumor progressing, recurring, or spreading systemically. None of the available investigations can accurately distinguish FTC from FA, FVPTC, and other follicular-patterned lesions of the thyroid gland. Follicular carcinoma is a complex and heterogeneous disease and is more aggressive than PTC. The risk stratification and treatment of FTC should therefore not be inferred from the experience gathered during the treatment of PTC, as FTC and PTC are clinically and genetically distinct diseases. Integration and simultaneous analysis of findings from the available omics are likely to lead to a more accurate diagnosis, prognostication, and prescription of effective treatment for FTC. The adoption and use of multi-omics and AI in the management of FTC should therefore be encouraged.

## Figures and Tables

**Figure 1 biomedicines-11-01217-f001:**
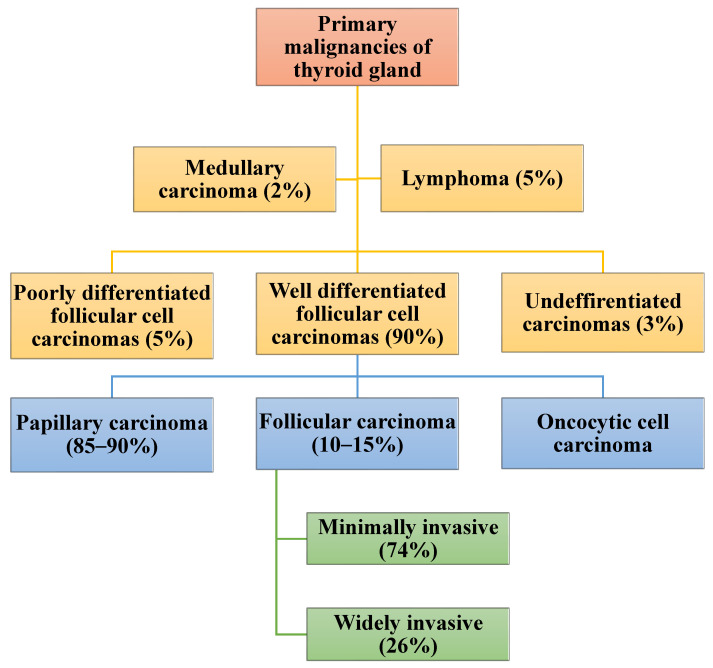
Classification and proportional rate of occurrence of follicular carcinoma and other thyroid malignancies.

**Figure 2 biomedicines-11-01217-f002:**
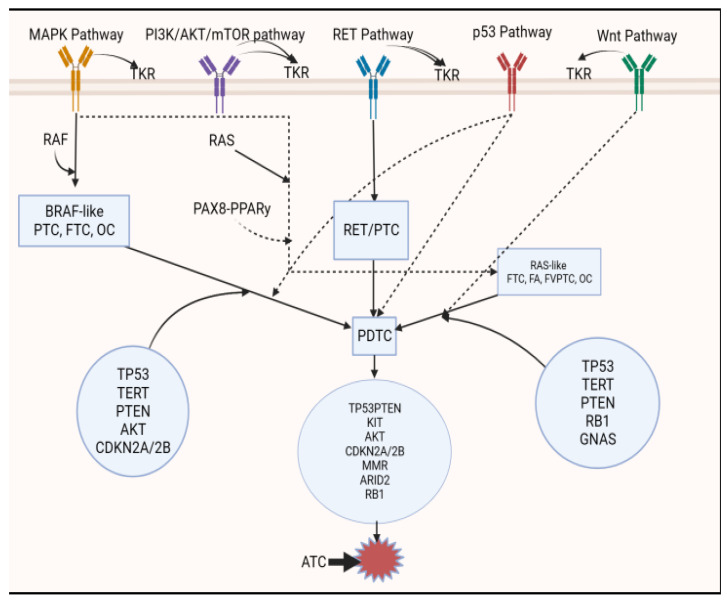
Pathways in the development, progression, metastasis, and de-differentiation of follicular neoplasms of the thyroid. The initiating event may be a gain mutation or re-arrangement in the genes that regulate the transmembrane tyrosine kinase receptors or involve intracellular kinases. The *MARK*, *PI3K/AKT/mTOR* and *RET* pathways are involved in the early phase of tumorigenesis. The *RAS* mutation is an upstream event for FTC, FA, FVPTC, and rarely PTC. *RAF* mutations are seen more frequently in PTC. The *p53, Wnt*, *PTEN*, *TERT,* and other mutations or rearrangements occur later as WDTC progresses towards PDTC or ATC. (Created using BioRender.com accessed on 13 February 2023).

**Figure 3 biomedicines-11-01217-f003:**
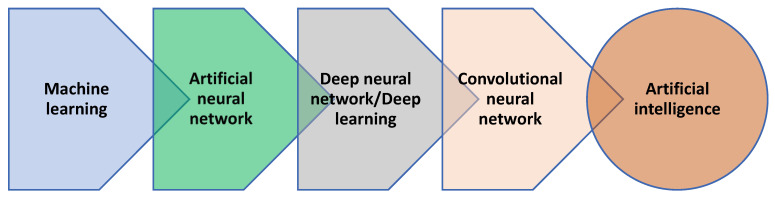
Progressive increase in the levels of complexity from machine learning to CNN in AI programs.

**Figure 4 biomedicines-11-01217-f004:**
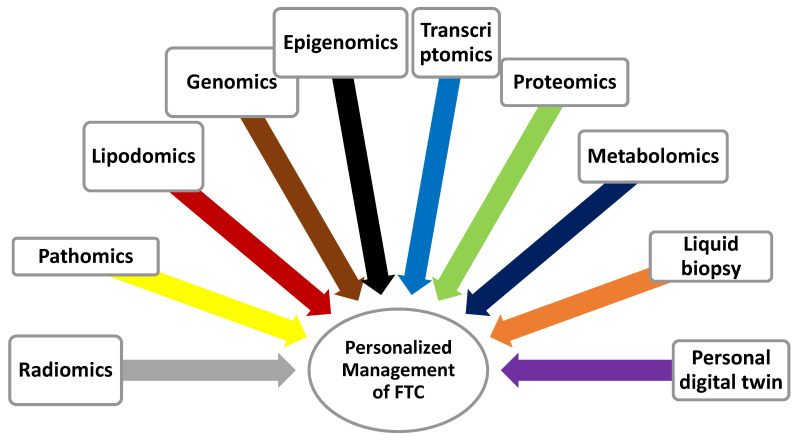
Components of the multi-omics that can be integrated and analyzed using the AI platform for computer-aided decision making in the management of FTC.

**Table 1 biomedicines-11-01217-t001:** Comparison of demography and clinicopathological features of follicular and papillary carcinomas.

Parameter	FTC	PTC
Gender predilection	Females	Females
Age [45,67]	>50 years	<50 years
Risk factors	Iodine deficiency, colloid goiter.	Radiation, iodine excess and thyroiditis.
Genetic predisposition	<3%	<5%
Tumour size at presentation [60]	Large	Small
Multifocality [68]	Rare	Common
FNAC diagnosis	No	Yes
Tumour capsule	Yes	No
Capsular and vascular invasion	Common	Rare
Main subtypes [69,70]	3	>12
Prevalent mutations or re-arrangements [71]	RAS, PAX8, PPARƴ, VEGFR.	BRAF, RET, VEGFR.
Lymph node metastasis [68,72]	Rare (<10%)	Common (20–90%)
Haematogenous metastasis [45,68]	Frequent (29%)	Rare (9%)
Standard surgical treatment	Lobectomy or total thyroidectomy	Lobectomy or total thyroidectomy with or without lymph node dissection.
10-Year disease free survival [60,68]	72%	92%

**Table 2 biomedicines-11-01217-t002:** Lists of reported mutations, histone modifications, and overexpressed non-coding RNA in follicular carcinoma of the thyroid.

Gene	Nature of Change	Direction	Implication	Associated Thyroid Tumour/s	Targeted Therapy/ Druggable
*BRAF* [35,63,166,185,186].	Mutation or re-arrangement	Over-expressed.	Inconclusive	PTC, FTC, FVPTC, PDTC, ATC	Yes
*RAS* [6,35,43,47,187].	Mutation or re-arrangement	Over-expressed.	Inconclusive	FTC, FA, FVPTC, PTC, PDTC, ATC	Yes
*PAX8/PPAR* *ƴ*	Fusion	Over-expressed	Favourable	FTC, FA, FVPTC, OC	Yes
*PAX8/PPARG* [9,116].	Rearrangement	Over-expressed	?	FTC, FA, FVPTC	
*RET* [3].	Fusion	Over-expressed	Aggressive	MTC, PTC	Yes
*TP53* [7,96,98,133,136].	Mutation	Down-regulated	Aggressive	PDTC, ATC	-
*TERT* [3,98,136,188].	Mutation	Over-expressed	Aggressive	FTC, PTC, PDTC, ATC	
*PTEN* [136,189].	Mutation	Down-regulated	Aggressive	PDTC, ATC	-
*E-Cadherin* [44,49].	Mutation	Down-regulated	Aggressive		.
*ß-Catenin*	Mutation	Down-regulated	Aggressive	PDTC, ATC	-
*TSHR* [133].	Mutation	Down-regulated	Aggressive	PDTC, ATC	-
*PIK3/AKT/mTOR* [9,136].	Mutation	Over-expressed	Aggressive	FTC, PDTC, ATC	Yes
*ALK* [9,98,168].	Mutation	Over-expressed	Aggressive	PDTC, ATC	Yes
*AKT1* [190,191].	Mutation	Over-expressed	Aggressive	FTC	-
*MEK* [9].	Mutation	Over-expressed	Aggressive	PTC, FTC	Yes
*ERK* [186].	Mutation	Over-expressed	Aggressive	PTC, FTC	Yes
*RB1* [160].	Mutation	Down-regulated	Aggressive	PTC, FTC, OC, ATC	Yes
*NF-1* [9,98,154].	Mutation	Over-expressed	Aggressive	PDTC, ATC	Yes
*CDKN2A/2B* [192].	Mutation	Over-expressed	Aggressive	PDTC, ATC	-
*ARID1A, ARID1B, ARID2, ARID5B* [98].	Mutation	Over-expressed	Aggressive	PDTC, ATC	-
*KIT* [9].	Mutation	Over-expressed	Aggressive	ATC	Yes
*RBM10* [193].	Mutation	Over-expressed	Aggressive	PTC, FTC, PDTC, ATC	-
*NTRK1* [9,98].	Mutation	Over-expressed	-	PTC, FTC	-
*EIF1AX* [9,98,114,194].	Mutation	Over-expressed	Aggressive	FA, OC, PTC, PDTC, ATC	-
*ATM* [78].	Mutation	Over-expressed	Aggressive	PDTC, ATC	-
*KMT* [194].	Mutation	Over-expressed	Aggressive	PDTC, ATC	-
*HMT* [98].	Mutation	Over-expressed	Aggressive	PDTC, ATC	-

*ALK*: alkaline phosphatase; *AKT*: Ak strain transforming; *ARID*: AT-rich interactive domain-containing protein 2; *ATM*: ataxia-telangiesctasia mutated; *CDKN2A*: cyclin-dependent kinase inhibitor 2A; *HMT*: histone methyltransferases; *mtDNA*: mitochondrial DNA 1; *NF1*: neurofibromin 1; *NTRK1*: neurotrophic tyrosine kinase receptor 1; *RB1*: retinoblastoma 1; *RBM10*: RNA-binding motif 10.

**Table 3 biomedicines-11-01217-t003:** A list of transmembrane receptors, including their ligands, whose expression is altered and plays a role in the tumorigenesis and progression of FTC and other TCs.

Receptor/Ligand	Expression	Implication	Availability of TK Inhibitor
Vascular endothelial growth factor receptor (VEGFR) [154,156,204].	Increased	Aggressive disease	Yes
Epidermal growth factor Receptor (EGF) [47].	Increased	Aggressive disease	Yes
Thyroid stimulating hormone receptor (TSHR)	Decreased	Aggressive disease	-
E-Cadherin [44].	Decreased	Aggressive disease	-
ß-Catenin [154]).	Increased	Aggressive disease	-
Galectin-3 [88,183,202].	Increased	Aggressive disease	-
Human bone marrow endothelial cell marker-1 (HBME-1) [110,183].	Increased	Unknown	-
Estrogen receptor α/β (ERα and ERβ) [205].	Increased	Aggressive disease	-
Human epidermal receptor 1 and 2 (HER1 and HER2) [154,205].	Increased	Aggressive disease	Yes
G protein-coupled estrogen receptor (GPER1) [191].	Increased	Aggressive disease	-
Fibroblasts growth factor receptor (FGFR) [204].	Increased	Aggressive disease	Yes
CXCR1 [206].	Increased	Aggressive disease	-
Thyroid transcription factor-1 (TFF-1) [75].	Decreased	Aggressive disease	-
Thyroglobulin	Decreased	Aggressive disease	-
Hepatocyte growth factor receptor (HGFR) [83].	Increased	Aggressive disease	Yes
Placental growth factor receptor (PIGFR) [204].	Increased	Aggressive disease	-
Platelet derived growth factor receptor (PDGFR) [154,204].	Increased	Aggressive disease	Yes
Angiopoietin-like protein 1 (ANGPTL1) [207].	Decreased	Aggressive disease	-
Somatostatin Receptor Type (SSTR2) [161].	Increased	Aggressive disease	Yes
Lysine-specific histone demethylase 1A [208].	Increased	Aggressive disease	Yes
Human bone marrow endothelial cell marker-1 (HBME-1) [110].	Increased	Unknown	-

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
