# Peer review of "Multi-Omics and Management of Follicular Carcinoma of the Thyroid"

_biomedicines, 2023, doi:10.3390/biomedicines11041217_

Round 1
Reviewer 1 Report
The manuscript entitled “Multi-omics and management of follicular carcinoma of the thyroid” by Luvhengo et al presents a detailed narrative on the difficulties encountered in the diagnosis, staging, risk assessment, management, and follow-up of thyroid cancer patients, and the potential of multi-omics for better decision-making in the management of follicular thyroid carcinoma (FTC). Authors state that the process of diagnosing, staging, assessing risk, managing, and following up with patients who have FTC follows a similar model to that of papillary thyroid carcinoma (PTC), despite FTC being more aggressive. They have also mentioned that FTC has a higher tendency for hematogenous metastasis compared to PTC, and it is a genetically and phenotypically diverse disease, and that the identification of markers of aggressive FTC relies on the proficiency and meticulousness of a pathologist during histopathological analysis. The authors mentioned that the use of various techniques such as radiomics, pathomics, genomics, transcriptomics, metabolomics, and liquid biopsy has improved understanding of thyroid cancer. Finally, the authors discussed the challenges encountered in diagnosing, staging, assessing risk, managing, and following up with thyroid cancer patients, and how multi-omics can improve decision-making in the management of follicular carcinoma.
Comments:
1. Please explain Tg.
2. In page 9, line 371, briefly provide details about side effects.
3. It would be better to have a table highlighting the differences between FTC and PTC.
4. Mention the gender-specific incidence of FTC, if the information is available.
5. In page 11, line 455, provide a list of tyrosine kinase inhibitors used in the treatment of FTC.
6. In page 11, line 466, provide a list of multi-kinase inhibitors used in the treatment of FTC.
7. In page 11, line 466 – Briefly explain thyroid cancer dosimetry and its use in the diagnosis of FTC.
8. In page 12, line 507 – Expand TERT.
9. In page 13, line 569 – provide examples of transmembrane driven mitogen activating kinases relevant to this review.
10. Figure 2 – Provide a detailed legend.
11. In page 15, line 616 – there is a typo
12. In section 2.1 under the heading “Pathomics in the diagnostic work-up of FTC, please explain in detail the upregulated and downregulated genes that regulate proliferation, differentiation, mobility and apoptosis.
13. In Table 2, the title says “Lists of reported mutations, histone modifications and over expressed non-coding RNA in follicular carcinoma of the thyroid”. However, it is not clear which gene has histone modifications or over expressed non-coding RNA. Please make it clearer.
14. In page 18 under the heading “2.1 Proteomics and metabolomics of TC”, there is no mention about the details of metabolomics relevant to this review. Please explain in detail.
15. In page 20, line 737, there is a typo – please correct.
16. There are some grammatical and spelling mistakes in the manuscript that needs to be corrected.
Author Response
Comments and Suggestions for Authors
|
Comments |
Response |
Action |
|
1. Please explain Tg. |
Accepted. |
Thyroglobulin added at initial use (See Paragraph 4 in the Introduction Section. |
|
2. In page 9, line 371, briefly provide details about side effects. |
Accepted. |
Details of side-effects have been added. a) Side effects of I-131 have been added Paragraph 2 in Subsection 2.4.2. b) Side effects of excessive TSH suppression have been added in Subsection 2.4.3. c) Side-effects of TKIs and MTIs have been added in Subsection 2.7. |
|
3. It would be better to have a table highlighting the differences between FTC and PTC. |
Accepted. |
A table comparing FTC and PTC has been added (see Table 1). The previous Table 1 which covered the AJCC Staging has been deleted as was requested by the other reviewer. |
|
4. Mention the gender-specific incidence of FTC, if the information is available. |
Accepted. |
A statement on the comparative rate of occurrence FTC in females and males has been added. See Table 1 and Paragraph 2 of Section 2. |
|
5. In page 11, line 455, provide tyrosine kinase inhibitors used in the treatment of FTC. |
Accepted. |
Examples of TKIs used in the treatment of FTC have been added. See Paragraph 3 of Subsection 2.5. |
|
6. In page 11, line 466, provide a list of multi-kinase inhibitors used in the treatment of FTC. |
Accepted. |
Examples of MKIs used in the treatment of FTC have been added. See Paragraph 3 of Subsection 2.5. |
|
7. In page 11, line 466 – Briefly explain thyroid cancer dosimetry and its use in the diagnosis of FTC. |
Accepted. |
A brief explanation of I-131 dosimetry and its benefits during treatment of thyroid cancer has been added. See Subsection 2.4.2. |
|
8. In page 12, line 507 – Expand TERT. |
Accepted. |
TERT has been expanded. See Subsection 2.5.1. |
|
9. In page 13, line 569 – provide examples of transmembrane driven mitogen activating kinases relevant to this review, |
Accepted. |
Examples of trans-membrane mitogen activating kinases have been added (See Paragraph 2 of Section 3. They are also depicted in Figure 2. |
|
10. Figure 2 – Provide a detailed legend. |
Accepted. |
A detailed legend has been added. See Figure 2 in Section 3. |
|
11. In page 15, line 616 – there is a typo. |
Accepted. |
The typo has been corrected. |
|
12. In section 2.1 under heading “Proteomics in the diagnostic work-up of FTC, please explain in detail the upregulated and downregulated genes that regulate proliferation, differentiation, mobility and apoptosis. |
Accepted. |
The section on proteomics has been expanded. See Subsection 4.4. The specific proteins and direction of change are included in Table 3. |
|
13. In Table 2, the title says “Lists of reported mutations, histone modifications and over expressed non-coding RNA in follicular carcinoma of the thyroid”. However, it is not which gene has histone modifications or over-expressed non-coding RNA. Please make it clearer. |
Accepted. |
We have separated the discussion of non-coding RNAs and Histone modifications is separated. See Subsections 4.3.1 and 4.3.2. |
|
14. In page 18 under the heading “2.1 Proteomics and metabolomics of TC”, there is no mention about the details of metabolomics relevant to this review. Please explain I detail. |
Accepted. |
Discussion of proteomics, metabolomics and liquid biopsy have separated into two subsections (See Subsections 4.4 - 4.6). |
|
15. In page 20, line 737, there is a typo – please correct. |
Accepted. |
The multiple typos have been corrected. |
|
16. There are some grammatical and spelling mistakes in the manuscript that needs to be corrected. |
Accepted. |
The manuscript was re-checked and grammatical and spelling mistakes throughout the text have been corrected. |

Reviewer 2 Report
According to its title, the submitted manuscript issue is the role of multi-omics in managing follicular thyroid cancer. However, in its current form, the paper does not cover the topic adequately.
The authors should focus on multi-omics. However, in the first part of the manuscript, they provide a very detailed description of the general management of thyroid carcinoma. It should be significantly shortened, and unnecessary tables (for example current AJCC staging system) and data should be deleted.
In the second part of the paper, the omics’ role is briefly and superficially discussed. The authors should focus on FTC (plus distinguishing the FTC from other cancers), whereas much information is given on thyroid cancer in general (particularly in the tables). Some data are unnecessary (for example, the information on the number of chromosome pairs in the human genome, etc.). Some are missing or are inadequately covered (for example, the role of molecular testing of thyroid nodules in FTC diagnosis). Frequently, omics-related techniques are mentioned as helpful and not discussed further (for example – in the paragraph on liquid biopsy (lines 697-704), it is stated that this technique provides a non-invasive approach to the management of FTC, and is helpful for differentiating FA from FTC, etc. – without any details given). No examples of AI adoption in FTC management are given, etc.
Therefore, I recommend the rejection of the manuscript in its current form and reconsidering its resubmission after it is rewritten to match the aim included in its title.
Author Response
Comments and Suggestions for Authors
|
Comments |
Response |
Action |
|
1. According to its title, the submitted manuscript issue is the role of multi-omics in managing follicular thyroid cancer. However, in its current form, the paper does not cover the topic adequately. |
We do not completely agree with the reviewer. a) Multi-omics is the consolidation and integration of various omics (Demographics, life-style, imaging, cytopathology and histopathology, genomics and proteomics which have been covered from section 2.1.
b) Section 4 highlights how the application of each of the omics can assist to address challenges which during diagnostic work-up, staging, risk stratification, treatment and follow-up that were identified in the preceding sections. |
Multi-omics and AI have been expanded. a) We have highlighted some of the challenges associated with the diagnostic work-up, staging, risk stratification, treatment and follow-up of patients in Section 2.
b) We have added more information various omics in Section 4. Each component of the omics is discussed separately. |
|
2. The authors should focus on multi-omics.
a) However, in the first part of the manuscript, they provide a very detailed description of the general management of thyroid carcinoma.
b) It should be significantly shortened, and unnecessary tables (for example current AJCC staging system) and data should be deleted. |
We do not completely agree with the reviewer. a) We feel a detailed discussion of the challenges which are encountered during evaluation of patients who have FTC is important as the application of various is meant to assist in addressing them.
b) We agree that the table containing the AJCC staging is not essential. |
a) We have deleted and left out information than is general and neither describing the challenges associated with FTC or comparing it with other tumours of the thyroid.
b) The table containing the AJCC staging has been deleted. The new Table 1 compares FTC with PTC as requested by the other reviewer. |
|
3. In the second part of the paper, the omics’ role is briefly and superficially discussed. a) The authors should focus on FTC (plus distinguishing the FTC from other cancers)., whereas such information is given on thyroid cancer in general (particularly in the tables).
b) Some data are unnecessary (for example, the information on the number of chromosome pairs in the human genome, etc.).
c) Some are missing or are inadequately covered (for example, the role of molecular testing of thyroid nodules in FTC diagnosis).
d) Frequently, omics-related techniques are mentioned as helpful and not discussed further (for example – in the paragraph on liquid biopsy (lines 697 – 704), it is stated that this technique provides a non-invasive approach to the management of FTC, and is helpful for differentiating FA from FTC, etc.
e) No examples of AI adoption in FTC management are given. |
We partially agree with the reviewer.
a) Agreed.
b) Agreed.
c) Agreed.
d) Agreed.
e) Agreed. |
a) A table comparing FTC and PTC has been added in the Introduction. See new Table 1).
b) Information on the number of chromosomes has been deleted. (See Section 3 Paragraph 1).
c) Specific information regarding mutational analysis, immunohistochemistry and spectrometry have been added.
d) A more detailed description of liquid biopsy and its role in FTC has been added (See Subsections 4.1-4.7).
e) Examples of AI adoption in the management of FTC have been added (See expanded Section 5). |
